# Effects of Metabolic Syndrome and Sex on Stress Coping Strategies in Individuals with Depressive Disorder

**DOI:** 10.3390/metabo13050652

**Published:** 2023-05-11

**Authors:** Eva Puchner, Martina Platzer, Nina Dalkner, Karin Schwalsberger, Melanie Lenger, Frederike T. Fellendorf, Human-Friedrich Unterrainer, Andreas Schwerdtfeger, Bernd Reininghaus, Eva Z. Reininghaus

**Affiliations:** 1Institute of Psychology, University of Graz, 8010 Graz, Austria; 2Department of Psychiatry and Psychotherapeutic Medicine, Medical University Graz, 8036 Graz, Austria; 3Institute of Religious Studies, University of Vienna, 1010 Vienna, Austria; 4Center for Integrative Addiction Research (CIAR), Grüner Kreis Society, 1110 Vienna, Austria; 5Faculty of Psychotherapy Science, Sigmund Freud University, 1020 Vienna, Austria

**Keywords:** depression, stress coping strategies, metabolic syndrome, sex, distraction strategy

## Abstract

Metabolic syndrome (MetS) is related to depression and contributes to reduced life expectancy in individuals with mental disorders. Stress coping strategies are important factors in the development and maintenance of depressive disorders and have been related to metabolic disturbances. The aim of this study was to determine whether there is a difference in the use of positive (re- and devaluation, distraction, and control) and negative stress coping strategies in relation to patients’ MetS. A sample of 363 individuals (*n* female = 204, *n* male = 159) with a diagnosis of depression was measured with the Stress Coping Style Questionnaire and the Beck Depression Inventory. In addition, we collected data on MetS (waist circumference, triglycerides, high-density lipoprotein, fasting glucose/diabetes, blood pressure/hypertonia) according to the International Diabetes Federation. A 2 × 2 design including Mets (with vs. without) and sex (female vs. male) was performed to test for differences in stress coping strategies. Individuals with depression and MetS scored higher on distraction strategies than depressed individuals without MetS (*p* < 0.01, corrected with false discovery rate). In addition, we found sex differences in stress coping strategies indicating that women with depression scored higher on distraction strategies (*p* < 0.001, FDR corrected), as well as negative strategies (*p* < 0.001, FDR corrected), than men. No significant interaction between MetS and sex was found regarding the higher value of stress coping strategies. Findings suggest that individuals with depression and MetS used distraction strategies to a higher amount to cope with stress, which could be stress eating in some cases, than those without MetS. Women with depressive disorders had higher values than men on other coping strategies in our sample of individuals with depression. A better understanding of MetS and sex-specific differences in stress coping strategies might help to plan more effective preventive strategies and personalized treatment options for depression.

## 1. Introduction

Depressive disorder is a major mental illness with a lifetime prevalence of 16.6% [1]. There is a sex gap, with women suffering from this disease about twice as often as men [2]. According to the World Health Organization (WHO) [3], depressive disorder is one of the world’s leading causes of disability. It is significantly linked to social and economic burdens for individuals, society, and relatives [3,4]. Furthermore, metabolic syndrome (MetS) is one of the comorbidities that frequently co-occur with depression and may become worse over the course of the disease [5].

MetS is defined by the International Diabetes Federation (IDF) as a combination of abdominal obesity, raised blood pressure/hypertension, raised fasting plasma glucose, raised triglycerides, and reduced high-density lipoprotein cholesterol (HDL) [6]. MetS is a risk state, which can lead to cardiovascular disease and type 2 diabetes [7], atherosclerosis, chronic kidney disease, hyperuricemia/gout, and obstructive sleep apnea [8]. The prevalence of MetS in the general population varies widely and depends on the definition used, age, sex, socioeconomic status, and ethnic background [7]. Using the IFD criteria, the European MetS prevalence was found in 38% of women and 41% of men [9]. One meta-analysis [10] even suggests that women with MetS are at higher risk for cardiovascular disease than men. In severe psychiatric disorders including schizophrenia and affective disorders, MetS is highly prevalent [11,12]. It has been found that MetS negatively affects cognitive performance in individuals with bipolar disorder [13], increases mortality [14], and impairs quality of life [15].

In depressive disorders, the prevalence of MetS is about 31% [12]. Women with depression are at higher risk of MetS as compared to men [16]. In addition, it could be shown that there is a link between MetS and increased depressive symptoms in women, but not in men [17]. In a study by Mulvahill et al. [18], depression accompanied by MetS in older adults was related to greater symptom severity, chronicity of depression, and poorer antidepressant response. In a sample of individuals with major depressive disorder combined with anxiety, it has been shown that individuals with MetS had higher scores for depression, anxiety, additional psychiatric symptoms, and suicide attempts in the past than those without MetS [19].

Stress and coping strategies are essential factors for mental and physical health [20]. In the development of psychiatric disorders such as depression and anxiety disorders, stress plays an important role [21]. According to the vulnerability stress model [22], the interaction of genetic, biological, and social vulnerability and acute/chronic stress is crucial for the development of depression. This process is influenced by psychosocial factors including lack of resilience, less social support, low education, and lacking stress coping strategies, which contribute to the acute and long-term consequences of depression [22]. In addition, evidence showed that chronic psychological and occupational stress also contributes to the development of MetS [23,24]. At the physiological level, it has been shown that stress is related to some of the parameters of MetS [25]. Perceived stress led to increased blood pressure, triglyceride levels, glucose levels, and decreased serum HDL [25]. However, stress may affect MetS as well through behavioral pathways. Especially chronic stress has been associated with unhealthy behavior such as reduced physical activity [26], higher food intake [27], and emotional eating [28]. As acute and chronic stress factors are not always avoidable, it is particularly important to take a closer look at individual stress management strategies when investigating the occurrence of MetS in people with depression on stress and stress coping.

Stress coping strategies are activating measures taken by individuals to prepare for, shorten, prevent, mitigate, end, or adapt to a stressful event [29]. Thus, stress coping styles can compromise behavioral, cognitive, emotional, and physiological processes and can be considered habitual characteristics that are relatively constant over time and independent of the type of stress [29]. In the literature, two coping categories have been repeatedly reported: problem-focused and emotion-focused [30]. Other authors cited a third strategy of coping: avoidant coping, which can be described as a denial of the problem [31]. Another important theoretical and practical coping construct that is used in this investigation is the division into positive/adaptive versus negative/maladaptive coping strategies [29]. Positive strategies include the attempt to distract oneself from stress, reevaluate or devalue stressors, control the situation and one’s reactions, or attribute coping skills to oneself. These strategies can be defined as stress-decreasing strategies, whereas negative strategies are stress-increasing strategies [32]. Negative strategies include not being able to mentally detach from the stressful situation, giving up with feelings of helplessness, wanting to escape the stressful situation, and blaming oneself [32].

Individuals differ greatly in terms of their stress coping strategies. In addition, numerous studies have shown that there are also sex differences in stress coping; however, research is inconclusive. Female individuals tend to use more maladaptive coping strategies and less adaptive coping than male individuals [32,33]. However, another study showed that women were more likely to use a larger variety of coping strategies than men [34].

Negative coping strategies were associated with depression [35,36]. Individuals with depression were found to often use escape tendencies, resignation strategies [37], denial, and behavioral disengagement (hopelessness) to cope with stress [38]. Research on depression also shows that women and men considerably differ in the stress coping strategies used. Rumination, a core symptom of depression, [39] has been found to be used more frequently by women than men [40]. Accordingly, women who used less often positive reframing showed higher scores in depression, which could not be demonstrated in men [41].

To date, the relationship between MetS and stress/stress coping strategies is not fully elucidated yet. There are indications that Pakistani immigrant women with MetS had lower scores in problem-solving coping and had associated higher depression scores than those without MetS [42]. Abdullah et al. [43] showed that patients with MetS who scored high on stress scales engaged mainly in negative coping strategies such as behavioral disengagement (escape), venting, and self-blame while individuals with MetS reporting low stress used mainly planning coping, such as thinking about a strategy to deal with stress. Furthermore, the greatest risk for MetS was found in people who had high scores in rumination [44]. However, there are some indications that there is a relationship between individual metabolic parameters and stress coping. Obese people tended to exhibit more passive and negative coping strategies including self-criticism, wishful thinking, and social withdrawal [28]. Moreover, patients with heart disease and hypertension used less appropriate stress coping strategies [45]. Another study found lower scores in adaptive coping (by positive self-instruction) in patients with low HDL cholesterol in comparison to patients with regular HDL cholesterol [46]. Furthermore, a recent study by Łukasiewicz et al. [47] found higher glucose levels in women with type 2 diabetes that used maladaptive stress coping strategies compared to a group that used adaptive strategies.

This investigation aims to fill a gap in the literature by exploring which stress coping strategies are used in individuals with a diagnosis of depression and MetS. Since sex differences and depressive symptoms appear to play a significant role in the use of coping strategies, the factor of sex and depressive symptoms were included in the analysis. We hypothesized that individuals with depression and MetS would have higher values on negative (stress-increasing) coping strategies, whereas individuals with depression without MetS would have higher values on positive (stress-decreasing) strategies. In addition, we hypothesized that female individuals with depression would have higher values on negative (stress-increasing) coping strategies than male individuals with depression.

## 2. Materials and Methods

### 2.1. Procedure

The data of the present study are part of a large-scale study on the neurobiological background of burnout symptoms in psychiatric disorders. The project was conducted in a psychiatric rehabilitation center in Austria between April 2015 and April 2016. All patients treated at the clinic and meeting the inclusion criteria were invited to participate. The treatment was scheduled for four or six weeks, and the treatment focus was affective and stress-related disorders after acute illness episodes. The inclusion criteria for the present study were a diagnosis of unipolar depression: F32 (depressive episode) or F33 (recurrent depressive disorder) according to the International Classification of Diseases (ICD-10) [48], confirmed by medical doctors and a complete data set with target parameters (MetS, stress coping). Exclusion criteria were a diagnosis of bipolar disorder, schizophrenia, or addictive disorder, moderate and severe intellectual impairment, severe brain disease, and acute psychotic symptomatology. Individuals were assigned to the MetS groups retrospectively, depending on whether they fulfilled the criteria of MetS or not. The investigation included blood, medical, and psychiatric history, anthropometric measures, psychometric tests, and cognitive performance tests.

The study protocol has been approved by the Ethics Committee of the Federal State Upper Austria (EK-number: 24-14) (in compliance with the current revision of the Declaration of Helsinki, ICH guideline for good clinical practice, and current regulations. All participants provided written informed consent prior to participation in this study.

From this existing data set, individuals who fulfilled all necessary criteria for this study were drawn (*n* = 363). An a priori power analysis with G × Power 3.9.1.7 indicated a minimum total sample size of *n* = 352, using effect size f^2^(V) = 0.025, α err prob = 0.05. Power (1-β err prob) = 0.95 for the global MANCOVA effects.

### 2.2. Psychological Inventories

The Stress Processing Questionnaire (SVF-78) [32] was used as a self-reported measure to assess different coping styles. It covers a total of 13 strategies with 6 items each, which are merged into 3 principal scales (see Table 1): positive, negative, and neutral strategies. The direction of coping strategies can be either stress decreasing (positive) or stress increasing (negative). Neutral strategies are not part of the negative or positive sub-scales, as they are dependent on the context of the situation and the personality. Furthermore, the positive strategies can be divided into 3 sub-areas: Positive 1 (Pos 1): re- and devaluation strategies, Positive 2 (Pos 2): distraction strategies, and Positive 3 (Pos 3): control strategies. The SVF-78 uses a 5-point Lickert scale ranging from 0 (not at all) to 4 (very likely) which indicates how likely an individual is to apply a certain coping strategy. For this present study, we used the raw score of the 3 positive subscales (re- and devaluation, distraction, and control strategies) and the negative scale. Thus, we used a total of 4 stress coping scales as dependent variables (3 positive and 1 negative strategy). The neutral strategy was not used for the analysis. Furthermore, the score of each strategy (3 positive and 1 negative strategy) was standardized as a T score. The reference is a non-clinical population aged 20–79 years, and the mean and standard deviation of the reference population correspond to 50 and 10 of the T score, respectively [32]. The average use of coping strategies is reported by a T score between 40 and 60. A T score lower than 40 indicates reduced use of a coping strategy, while a score higher than 60 indicates higher use of a coping strategy.

Depressive symptoms were assessed with the Beck Depression Inventory (BDI-II) [49]. The depressive symptoms were recorded, as they can influence stress coping strategies. The self-administered questionnaire consists of 21 items, which are scored on a scale of 0 to 3. A BDI-II score of 14–19 indicates mild depressive symptoms, 20–28 moderate depressive symptoms, and 29–63 severe depressive symptoms.

### 2.3. Physiological Assessment

Anthropometric and fasting biochemical examinations were collected. MetS was defined according to the IDF criteria [6] as a cluster of high-risk factors: central obesity (waist circumference for men ≥ 94 cm or women ≥ 80 cm) plus co-occurrence of at least two of the following factors:
Serum triglycerides levels ≥ 150 mg/dL (or treatment for hyperlipidemia);HDL cholesterol < 40 mg/dL in men or <50 mg/dL in women;Blood pressure ≥ 130/85 mmHg (or diagnosed hypertension);Fasting blood glucose ≥ 100 mg/dL (or presence of Type 2 diabetes).

### 2.4. Statistical Analyses

To attest to potential differences in the use of different coping strategies between MetS (with vs. without) and sex (female vs. male), a two-way multivariate analysis of covariance (MANCOVA) was used. As dependent variables, we used the three positive sub-scales (Pos 1: re- and devaluation, Pos 2: distraction, Pos 3: control) as well as the negative sub-scale of the SVF-78. Thus, we used a total of 4 stress coping scales as dependent variables (3 positive and 1 negative strategy). Covariates included age, level of education, and BDI-II score. We used raw scores for the MANCOVA analysis.

Before conducting the MANCOVA, the prerequisites were checked. Normal distribution was assessed by the Shapiro–Wilk test (*p* > 0.05); if not achieved, the MANCOVA was still calculated, as it is relatively robust to violations of the normal distribution [50]. Levene’s test showed homogeneity of variances for all dependent variables (*p* > 0.05) and homogeneity of covariances was tested by Box’s M Test (*p* > 0.05).

For the differences in anamnestic and clinical characteristics, we used Chi-square tests (for nominal data) and t-tests (for categorical data). Furthermore, the score of each strategy (3 positive and negative strategies) was standardized as a T score.

The statistical analyses were performed using the German version of SPSS 27. The alpha level of the statistical analyses was set at *p* < 0.05. We applied the false discovery rate (FDR) to correct for multiple testing (corrected significance level = 0.0125) [51].

## 3. Results

### 3.1. Descriptive Statistics

A total of 363 individuals with unipolar depression were included in the study: 204 females (56.2%) and 159 males (43.8%). The average age of the participants was 53 years (*SD* = 7.17) with a range from 22 to 72 years, and 117 (32.2%) individuals fulfilled the criteria of MetS at the time of testing. Participants showed a moderate severity of depression (BDI-II: *Mean* = 20.09, *SD* = 10.29) and an average body mass index (BMI) of 27.2 kg/m^2^ (*SD* = 5.44). In this sample, 156 individuals had low education (43%), indicating that they had not completed high school, and 207 individuals had a high school degree (57.0%). Regarding ICD-10 diagnosis, 146 (40.2%) individuals had a diagnosis of a single depressive episode (F32) and 217 (59.8%) had a diagnosis of recurrent depressive disorder (F33).

Individuals with depression showed the highest mean T-scores in negative strategies (*Mean* = 58.69, *SD* = 10.05), followed by distraction strategies (Pos 2) (*Mean* = 48.13, *SD* = 8.30), control strategies (Pos 3) (*Mean* = 45.43, *SD* = 9.76), and devaluation and revaluation strategies (Pos 1) (*Mean* = 44.56, *SD* = 8.98). The mean T-scores of all coping strategies are within the average range (40–60) according to the manual [32], indicating that the strategies used were at an average level. However, the mean T-score of the negative strategies was at the limit of the upper average range (60), suggesting the use of negative coping strategies to a higher amount in individuals with depression.

### 3.2. Chi-Square Test and t-Test

Table 2 presents differences in anamnestic and clinical data between MetS (with vs. without) and sex (female vs. male). Group comparisons revealed there was no difference between MetS (with vs. without) in BDI-II score, age, and education. In addition, female individuals were of a higher age and had a higher education than male individuals, while male individuals had a higher BMI than female individuals. No difference between sex in BDI-II score was found.

### 3.3. Multivariate Effects

The two-way MANCOVA (see Table 3) comparing MetS (with vs. without) and sex (female vs. male) in the use of different coping strategies showed a significant main effect MetS [*F*(4353) = 2.496, *p* < 0.05, *η_p_*^2^ = 0.028, *Wilks’ Λ* = 0.927] and a significant main effect sex [*F*(4353) = 8.875, *p* < 0.001, *η_p_*^2^ = 0.091, *Wilks’ Λ* = 0.909]. The interaction MetS by sex on the combined dependent variables was not significant [*F*(4353) = 1.845, *p* = 0.12, *η_p_*^2^ = 0.020, *Wilks’ Λ* = 0.980]. BDI-II score was a significant confounder [*F*(4353) = 43.982, *p* < 0.001, *η_p_*^2^ = 0.333, *Wilks’ Λ* = 0.667]. Age [*F*(4353) = 1.993, *p* = 0.10, *η_p_*^2^ = 0.022, *Wilks’ Λ* = 0.978] and education [*F*(4353) = 0.910, *p* = 0.46, *η_p_*^2^ = 0.010, *Wilks’ Λ* = 0.990] had no effect. Means and standard deviations are listed in Table 2.

### 3.4. ANCOVAs

ANCOVAs showed that MetS groups (with vs. without) differed in distraction strategies. In addition, there was a statistically significant difference between sex (female vs. male) for re- and devaluation strategies (Pos 1) [*F*(1356) = 3.912, *p* < 0.05, *η_p_*^2^ = 0.011], distraction strategies (Pos 2) [*F*(1356) = 12.260, *p* < 0.001, *η_p_*^2^ = 0.033], and negative strategies [*F*(1356) = 21.190, *p* < 0.001, *η_p_*^2^ = 0.056]. No significant interaction MetS by sex for any stress coping strategy was observed. The ANCOVAs are presented in Table 4.

FDR-corrected post hoc tests indicated that individuals with MetS had a significantly higher score in distraction strategies than individuals without MetS. Furthermore, female individuals scored significantly higher on distraction (Pos 2) and negative strategies than male individuals. The significant difference between sex (female vs. male) for re- and devaluation strategies (Pos 1) did not remain significant after FDR correction.

## 4. Discussion

This study aimed to determine differences in the use of positive and negative stress coping strategies between MetS (with vs. without) and sex (female vs. male) in individuals with a diagnosis of depression. Since depressive symptoms seem to play a significant role in the use of coping strategies, all analyses were corrected for the BDI-II score. Our findings show that individuals with depression and MetS had higher values on distraction strategies than individuals without MetS, indicating that individuals with depression and MetS tend to distract themselves from stress by turning to positive activities, which could be eating in some cases. In addition, we found that women with depression had higher values on distraction strategies and significantly higher values on negative strategies than men. These effects were independent of MetS. In the following, we will discuss our results in more detail.

First, in contrast to our expectations, no difference in negative strategies was found between individuals with depression and MetS compared to those without MetS. Our result is in line with the previous study, which did not find any difference between Pakistani immigrant women with and without MetS in depressive response patterns, such as being overwhelmed by the problem and ruminating about it, when coping with stress [42]. To date, only a few studies examined stress coping differences in individuals with and without MetS. Previous studies analyzing the effects of single metabolic parameters on stress coping strategies observed that some MetS parameters have been associated with negative strategies. Patients with hypertension and heart disease used less convenient stress coping strategies [45]. Obese people used more self-criticism and social withdrawal when coping with stress [28]. Another study found lower scores in positive self-instruction in patients with low HDL cholesterol compared to patients with regular HDL cholesterol [46]. It might be possible that only single MetS parameters have an impact on negative stress coping strategies, but not the full MetS diagnosis, which should be taken into account in further studies.

Although all participants in our study had a diagnosis of depression, the mean T-scores of the different stress coping strategies were within the average range. However, negative strategies were at the limit of the upper average range, suggesting the use of negative coping strategies to a higher amount in individuals with depression. This is in line with Cairns and colleagues [35] suggesting that negative coping strategies were associated with higher levels of depression. Furthermore, another study showed that individuals with depression were mainly using negative strategies such as resignation and escape tendencies [37]. Our result might indicate that MetS no longer has an impact on negative strategies in individuals with depression when negative strategies are already increasingly used, suggesting a ceiling effect in the use of negative coping strategies (at a T-score of 60).

Contrary to our assumption, the current results showed that individuals with depression and MetS had higher values on distraction strategies, one of the positive strategies, than those without MetS. The distraction strategy sub-scale (Pos2) in the SVF-78 questionnaire mainly consists of questions regarding distracting oneself from stressful situations or turning to other positive activities by focusing on something else, pursuing another activity, watching TV, or looking for something that makes one feel good. Furthermore, using food to face stress could be one distraction strategy and is quite common in individuals with overweight [52], and there is evidence that individuals tend to eat more unhealthy food under stress [27]. In addition, previous studies showed that obese individuals tended to have higher scores on emotional eating [28]. In hypertensive patients with MetS, it has been observed that depressive symptoms were related to an unhealthy diet and higher energy intake [53], leading to the assumption that individuals with MetS and depression would eat more unhealthy food when they are stressed. This is in line with the high self-reported scores on distraction strategies in our sample with MetS, assuming that food intake is one of the distraction strategies. However, according to the SVF-78, distraction strategies are among the positive strategies describing stress-reducing techniques. However, it is likely that distraction—when stress eating is practiced or high-caloric food serves as reward—can otherwise have negative effects on people with overweight or MetS [28].

In line with the results of this investigation, Yancura et al. [54] found an inverse association between positive coping and MetS in older men. The authors proposed that this relationship was mediated by emotion regulation strategies, which may exert protective effects. It is possible that there are other mediating and moderating factors such as self-efficacy [55], personality traits [56,57], or social resources [58], and even neurochemical processes [59] are implicated.

In accordance with expectations, our findings suggest that stress coping strategies in individuals with depressive disorders vary by sex. First, women with depression had higher values on distraction strategies compared to men with depression. Current literature indicates that engaging in gambling, smoking, drinking alcohol, and leisure activities/sports were more frequently found in men, whereas eating and going shopping were more common in women when distracting from stress [60]. Since the SVF-78 questionnaire did not focus on coping strategies preferably used by men, such as consuming psychotropic substances, alcohol, nicotine, and physical activity [60], it would be interesting to consider these coping strategies in future studies when investigating causing factors of MetS [61,62,63]. Our results are supported by Zellner et al. [64] who examined sex differences in eating behavior under stress and found more women than men reporting more stress eating. Second, we observed that women with depression had higher values on negative strategies than male individuals with depression, thus supporting previous results [32,33]. According to the authors of the SVF-78, individuals scoring high on negative strategies tend to escape from stressful situations, not being able to detach themselves mentally, give up with feelings of helplessness, and attribute burdens to their actions [32]. Our result is in line with other others showing more rumination and self-accusation in women than in men [40,65].

Interactions between MetS and sex were neither shown for re- and devaluation, distraction, control, nor for negative strategies. It seems that both are independent concepts related to stress coping strategies in individuals with depression.

### 4.1. Strengths and Limitations

The greatest strength of this study is the innovative study design and the large and homogeneous sample of well-diagnosed individuals with depression. However, there are also some limitations. First, no causal direction can be determined due to the cross-sectional design of the study. Hence, we do not know whether MetS has caused distraction strategies or distractive stress coping strategies have led to MetS. In addition, distraction has been clustered within the positive strategy scales, whereas we found it increased in individuals with depression and MetS. We can only assume that individuals with depression and MetS use higher and unhealthier food intake as one distraction strategy, though evidence for this assumption would have to be proved in further studies. Furthermore, other variables may have mediated the association (e.g., genetic or other dispositional factors such as personality style). Second, we did not monitor the medication intake of the participants, as there was a high variance in medication. Of note is that MetS has been related to the use of some psychopharmacological medications [66]. In addition, the model was not controlled for factors that may have influenced the results, such as time since the diagnosis of depression. Furthermore, other confounding factors, which may have potentially influenced the association between MetS and stress coping strategies, were not controlled for. Due to these factors, the interpretation of the study may be limited. Moreover, the severity of depressive symptoms was self-reported. In addition, the number of participants varied considerably between the groups, including a larger number of individuals without MetS (67.8%). There might be a potential inverse dependence between negative and positive strategies that may result in reduced statistical power. However, we checked this by also including inverse T scores in the MANCOVA model, which did not change the results. Furthermore, we have to take into consideration that research on differences in stress coping strategies is not conclusive, as outcomes appear to depend on how coping and stress coping strategies are defined and which coping instruments have been used [67]. Since the severity of depression was mild to moderate and not all patients suffer from acute depression at the time of testing, further research should be conducted in acute episodes, chronicity, severity, subtypes, and as well as in individuals with heightened suicide risk. In addition, future studies should investigate the impact of the single MetS parameters on stress coping strategies, to identify which metabolic parameters are related to which stress coping strategies and should be targeted in treatments.

### 4.2. Implications

A possible implication of this investigation could be to promote alternative stress coping strategies in individuals with depressive disorder. Individuals with MetS should learn about the variety of positive coping and strengthen those stress-reducing strategies that have nothing to do with distraction, but act more on a cognitive level (i.e., re- and devaluation and control strategies), so that distraction, i.e., including food intake, is needed less. Since there are differences between women and men in stress coping strategies in individuals with depressive disorder, the interventions should be adapted to sex.

## 5. Conclusions

This is the first study comparing the combined effects of MetS and sex on stress coping strategies in individuals with depression. Our findings showed that coping strategies in individuals with depression varied by sex, indicating that women with depression had higher values on negative and distraction strategies than men. Furthermore, individuals with MetS had higher values on distraction strategies than those without MetS. In light of our results, it is important to encourage the practice of other positive strategies in patients with depression, notably the de- and revaluation of control strategies, especially when they are overweight or have a diagnosis of MetS. In addition, special attention should be paid to what and how much is being eaten, why people need a distraction, and in which situation. Better stress management could also contribute to metabolic improvement. A better understanding of MetS and sex-specific differences in stress coping strategies might help to plan more effective preventive strategies and personalized treatment options for depression. Further studies are needed to validate and replicate the results of this investigation.

## Figures and Tables

**Table 1 metabolites-13-00652-t001:** SVF-78 The Stress Processing Questionnaire [32].

Positive strategies	Pos 1:	Re- and devaluation strategies	Under-evaluation	Attribute lower stress to oneself compared to others
			Guilt and denial	Emphasize lack of personal responsibility
	Pos 2:	Distraction strategies	Distraction	Distract yourself from stress-related activities and situations
			Alternative satisfaction	Turn to positive activities
	Pos 3:	Control strategies	Situation control	Analyze the situation, plan and execute problem solving
			Response control	Get your own reaction under control
			Positive self-instruction	Assure oneself of competence and ability to control the situation
Negative strategies			Escape	Tendency to escape a stressful situation
			Thought continuation	Rumination/cannot detach yourself mentally
			Resignation	Giving up with feelings of hopelessness and helplessness
			Self-blame	Attribute the burdens to their own wrong actions
Neutral strategies			Need for social support	Seek social support and help
			Active avoidance	Decide to prevent or avoid stress

Note. Pos 1 = positive scale 1: “devaluation and revaluation strategies”, Pos 2 = positive scale 2: “distraction strategies”, Pos 3 = positive scale 3: “control strategies”.

**Table 2 metabolites-13-00652-t002:** Differences in anamnestic characteristics and stress coping scores between MetS (with vs. without) and sex (female vs. male).

	With MetS		Without MetS		Differences between			Differences between		
	Female	Male	Female	Male	Mets (with vs. without)			Sex (Female vs. Male)		
	(*N* = 57)	(*N* = 60)	(*N* = 147)	(*N* = 99)	*t/χ*2	*df*	*p*	*t/χ*2	*df*	*p*
**Anamnestic data**										
Age M ± SD	54 ± 6.56	52 ± 7.24	54 ± 7.00	51 ± 7.52	−0.595	361	0.55	2.680	361	<0.01
BDI-II M ± SD	20.95 ± 10.15	19.80 ± 11.03	20.52 ± 9.71	19.12 ± 10.80	−0.345	361	0.73	1.163	361	0.25
BMI M ± SD	31.4 ± 5.41	31.4 ± 4.63	24.8 ± 4.99	25.7 ± 3.30	−11.493	204.267	<0.001	−2.195	360.604	<0.05
**Obesity**										
Yes *N* (%)	57 (48.7%)	60 (51.3%)	66 (70.2%)	28 (29.8%)	124.371	1	<0.001	0.899	1	0.34
No *N* (%)	0	0	81 (53.3 %)	71 (46.7%)						
**Education**										
Low education *N* (%)	25 (6.89)	29 (7.99)	48 (13.22)	54 (14.88)	0.712	1	0.40	9.827	1	<0.01
High education *N* (%)	32 (8.82)	31 (8.54)	99 (27.27)	45 (12.40)						
**ICD−10 Diagnosis**										
F32 *N* (%)	16 (4.41)	28 (7.71)	54 (14.88)	48 (13.22)	0.490	1	0.48	6.758	1	< 0.01
F33 *N* (%)	41 (11.29)	32 (8.82)	93 (25.62)	51 (14.05)						
**Stress coping scales** M ± SD										
Re- and devaluation (Pos 1)	8.64 ± 3.23	9.40 ± 3.98	8.01 ± 3.34	8.80 ± 3.32	−1.826	361	0.07	−2.324	361	<0.05
Distraction (Pos 2)	12.02 ± 3.58	9.81 ± 3.63	10.14 ± 3.89	9.43 ± 3.58	−2.436	361	<0.05	2.745	361	<0.01
Control (Pos 3)	14.78 ± 2.74	13.81 ± 3.68	14.18 ± 3.20	14.18 ± 3.49	−0.277	361	0.39	0.869	361	0.40
Negative	15.54 ± 4.29	12.79 ± 5.10	15.20 ± 3.72	13.66 ± 4.46	0.865	195.527	0.39	4.231	302.783	<0.001

Note. *N* = Number of individuals, SD = standard deviation; M = mean; % = percentage; MetS = metabolic syndrome; BDI-II = Beck Depression Inventory; BMI = body mass index; obesity = waist circumference for men ≥ 94 cm or women ≥ 80 cm; ICD−10 = International Classification of Diseases; F32 = depressive episode; F33 = recurrent depressive disorder; Pos 1 = positive scale 1: “devaluation and revaluation strategies”; Pos 2 = positive scale 2: “distraction strategies”; Pos 3 = positive scale 3: “control strategies”.

**Table 3 metabolites-13-00652-t003:** Two-way MANCOVA.

	*F*	*p*	*η_p_* ^2^
Main effect MetS	2.496	<0.05	0.028
Main effect sex	8.875	<0.001	0.091
Interaction MetS by sex	1.845	0.12	0.020
Age	1.993	0.10	0.022
Education	0.910	0.46	0.010
BDI-II score	43.982	<0.001	0.333

Note. MetS = metabolic syndrome, BDI-II score = Beck Depression Inventory, MetS (with vs. without) and sex (female vs. male) as independent variables, stress coping strategies (re- and devaluation, distraction, control and negative) as dependent variables, and controlling for age, education and BDI-II score.

**Table 4 metabolites-13-00652-t004:** ANCOVAs following two-way MANCOVA.

	Main Effect			Main Effect			Interaction		
	MetS			Sex			MetS by Sex		
	*F*	*p*	*η_p_* ^2^	*F*	*p*	*η_p_* ^2^	*F*	*p*	*η_p_* ^2^
**Stress Coping Strategies**									
Re- and devaluation (Pos 1)	2.764	0.10	0.008	3.912	<0.05	0.011	0.000	0.99	0.000
Distraction (Pos 2)	7.634	<0.01	0.021	12.260	<0.001	0.033	3.317	0.07	0.009
Control (Pos 3)	0.173	0.68	0.000	1.823	0.18	0.005	1.976	0.16	0.006
Negative	0.792	0.37	0.002	21.190	<0.001	0.056	2.600	0.11	0.007

Note. MetS = metabolic syndrome, Pos 1 = positive scale 1: “de- and revaluation strategies”, Pos 2 = positive scale 2: “distraction strategies”, Pos 3 = positive scale 3: “control strategies”, MetS (with vs. without) and sex (female vs. male) as independent variables, stress coping strategies (re- and devaluation, distraction, control and negative) as dependent variables, and controlling for age, education and BDI-II score.

## Data Availability

The data presented in this study are available on request from the corresponding author. The data are not publicly available due to privacy.

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
