# Peer review of "Effects of Metabolic Syndrome and Sex on Stress Coping Strategies in Individuals with Depressive Disorder"

_metabolites, 2023, doi:10.3390/metabo13050652_

Round 1

Reviewer 1 Report

This article submitted to the journal ‘Metabolites’ investigated which type of stress coping strategies used by the depression inpatients was associated with metabolic syndrome, and distraction, one of positive strategies was used by the subjects with metabolic syndrome more than those without metabolic syndrome. They also elucidated sex differences of using stress coping strategies. The analysis concept was unique, and interesting for readers who engages in a psychiatric field. However, the aim of the journal ‘Metabolites’ is that “Metabolites publishes original research articles and review articles in aspects of metabolites and metabolism relevant to the fields of metabolomics, metabolic biochemistry, computational and systems biology, biotechnology and medicine.“ This article does not provide information about metabolites or metabolomics, and cannot suggest an idea relevant to metabolites. Moreover, the quality of the manuscript has some problems. The review comments some concerns in short.

1.      The focus of this study is ambiguous. In the Abstract in L27, “A better understanding of MetS and sex-specific differences in stress coping strategies might help to plan more effective preventive strategies and personalized treatment options for depression.” This is not mentioned in the Conclusion in L388, and not related to metabolites.

2.      Positive (stress-decreasing), and negative (stress-increasing) strategies used in the subjects were assessed using the Stress Processing Questionnaire (SVF-78). The scoring is not clarified. Two strategies groups have inverse direction. In MOCOVA analysis, when the dependent variables with different directions are involved, the results may be unstable. In addition, the reference for T scoring in the manual is unknown (L187). What population is used for the standards? In L211, T values of three positive strategies are less than 50, and that of negative strategy is higher than 50. What do these scores indicate of positive and negative strategies? Box plots is better to depict the distribution. L300 the normal range of stress coping strategies should be described in the methods. Which is at the limit of the upper normal range (L301), mean, median, quartile, minimum, or maximum?

3.      L313 the information “buying something nice and eating good food” is not presented in the methods. Does Distraction strategy consist only buying and eating?

4.      L317-L325, “In hypertensive patients with MetS, it has been observed that depressive symptoms were related to an unhealthy diet and higher caloric and excessive cholesterol intake [47], leading to the assumption that individuals with MetS and depression could eat more unhealthy food when they are stressed. This could be a reasonable explanation for the finding that individuals with MetS scored higher on distraction strategies. In this study, the distraction strategy was part of the positive strategies, as it is stress-reducing, however, it may cause negative effects for people with MetS otherwise. Since unhealthy eating behavior led to the maintenance and 324 encouragement of overweight and obesity [26].” This indicates that the stress strategy “Distraction” as a cause leads metabolic syndrome as outcome. However, the model of MANCOVA hypothesized that metabolic syndrome brings stress coping strategies with confounders such as sex, depressive symptoms, and others. Description of results were not accurate.

5.      In the Discussion, metabolites or metabolism do not appear.

Minor points

6.      L172, In the definition of metabolic syndrome, “plus co-occurrence of two of the following factors” may be “at least two”

7.      L245 Mets -> MetS as seen elsewhere.

8.      L248, L259, x is replaced by “multiply character”.

9.      L280 adadditional

10.   L300 comma after “although” is not needed.

Author Response

Thank you for your precious comments and feedback on our paper.

Please find attached a point-to-point response and all changes are highlighted in the text marked with track changes. In addition, our senior authors proof-red the manuscript again.

We would like to express our thanks for your time and the questions raised which have been helped us to improve our manuscript significantly.

Reviewer 2 Report

1. Please provide a sample size calculation.

2. Line 172, for waist, instead > it should be >=.

3. Why Bonferroni correction was used? It is the weakest method according to the power of the test. Why f.e. Benjamini-Hochberg correction was not used? I strongly suggest recalculating the results.

4. Why authors did not assess the personality type that has a strong influence on coping with stress? It is a very important cofounder here.

5. Please provide time since the diagnosis of depression and used it in the analysis, as it can be a confounder in stress coping strategies.

6. Did the authors take into account the use of the Hamilton scale, apart from Beck's inventory? The Hamilton scale encloses also data regarding anxiety disorders.

7. Giving the descriptive statistics for age, round it into an integer, for BMI to one decimal place.

8. Figure 1 should be presented either as Box-plot or mean with a 95% confidence interval, depending on the data distribution. Moreover, it should be divided according to sex and MS. Add on the figure p-values for comparison between groups.

9. What test was used to assess the normality of the data?

10. In the article please use p-values rounded into two decimal places in case of non-significant results, and use p < 0.05, p< 0.01, and p < 0.001 options.

11. In tables 2 and 3 please use the rules as in p. 10. Please remove */**/***. Do not use bold with p values.

12. Please add in table 2 N(%) of subjects with overweight and obesity and compare between groups.

13. Did the authors use log-linear analysis to compare nominal data?

14. Please reconsider reanalysis and changes in the discussion according to mentioned above doubts.

15. Please refer to articles:

Orzechowska, A.; Bliźniewska-Kowalska, K.; Gałecki, P.; Szulc, A.; Płaza, O.; Su, K.-P.; Georgescu, D.; Gałecka, M. Ways of Coping with Stress among Patients with Depressive Disorders. J. Clin. Med. 202211, 6500. https://doi.org/10.3390/jcm11216500

Zhong, Y., Hu, M., Wang, Q. et al. The prevalence and related factors of metabolic syndrome in outpatients with first-episode drug-naive major depression comorbid with anxiety. Sci Rep 11, 3324 (2021). https://doi.org/10.1038/s41598-021-81653-2

Imaizumi, T., Toda, T., Maekawa, M. et al. Identifying high-risk population of depression: association between metabolic syndrome and depression using a health checkup and claims database. Sci Rep 12, 18577 (2022). https://doi.org/10.1038/s41598-022-22048-9

Limon VM, Lee M, Gonzalez B, Choh AC, Czerwinski SA. The impact of metabolic syndrome on mental health-related quality of life and depressive symptoms. Qual Life Res. 2020 Aug;29(8):2063-2072. doi: 10.1007/s11136-020-02479-5. Epub 2020 Mar 25. PMID: 32215841; PMCID: PMC7513573.

Kelly MM, Tyrka AR, Price LH, Carpenter LL. Sex differences in the use of coping strategies: predictors of anxiety and depressive symptoms. Depress Anxiety. 2008;25(10):839-46. doi: 10.1002/da.20341. PMID: 17603810; PMCID: PMC4469465.

Author Response

Thank you for your precious comments and feedback on our paper. We were able to address all the points you´ve mentioned. Below you find the point-to-point response and all changes are highlighted in the text marked with track changes. In addition, our senior authors proof-red the manuscript again.

We would like to express our thanks for your time and the questions raised which have been helped us to improve our manuscript significantly.

Reviewer 3 Report

This study investigates the relationship between MetS, depression, and stress coping strategies. MetS is a cluster of metabolic disturbances that increases the risk of cardiovascular disease, T2DM, and reduced life expectancy. Depression is a common mental disorder that affects many people worldwide, and MetS is related to depression and contributes to reduced life expectancy in individuals with mental disorders. Understanding the relationship between MetS, depression, and stress coping strategies can help develop effective preventive strategies and personalized treatment options for depression. The study found that individuals with depression and MetS used more distraction strategies than depressed individuals without MetS. This suggests that people with MetS may use distraction strategies to cope with stress. The study also found sex differences in stress coping strategies, with women with depression using more distraction and negative strategies and men with depression. These findings suggest that there may be differences in how men and women cope with stress in the context of depression. This can help develop more effective prevention and treatment strategies for depression, tailored to individuals' needs and circumstances. Overall, while the study provides valuable insights into the relationship between MetS, depression, and stress coping strategies.

Author Response

Thank you for your comments and feedback on our paper.

All changes are highlighted in the text marked with track changes. In addition, our senior authors proof-red the manuscript again.

We would like to express our thanks for your time and the questions raised which have been helped us to improve our manuscript significantly.

Reviewer 4 Report

The purpose of this study was to evaluate the effects of MetS and sex on stress coping strategies among individuals with depressive disorder.

The main questions of this study are as follows:

MetS is a “risk condition” rather than an actual “disease”. This risk condition may influence health risk and cardiovascular disease progression, rather than behavioral coping strategies. This is also the reason why few researchers have explored the effect of MetS on stress coping strategies. This study found that MetS was related to stress coping strategy comes from other factors that coexist with MetS, such as, the patient’s own stress status, individual personality traits, etc. However, this study did not assess the influence of these factors on the results. In addition, the analysis of this study did not control for confounding effects well, and the results of the study were not sufficient to support the research conclusions proposed by the authors.

This study also has the following issues to be clarified:

1.      Abstract. There was no numerical evidence in the abstract to support the conclusions of this investigation.

2.      Lines 39-55. The description about MetS is too cumbersome.

3.      Lines 56-72. This paragraph of statement was not related to the issues that this study want to investigate. The research question was the effect of MetS on stress coping strategies, but not the effect of stress on MetS.

4.      Lines 88-89. The reference was not provided to support the statement.

5.      Lines 130-147. The study did not give methods of participant selection, nor information about the setting, study dates, and periods of recruitment.

6.      Lines 169-177. When the MetS condition for individuals with depressive disorder was diagnosed? Did the patients recognize they had MetS condition?

7.      When the MetS condition was diagnosed in depressed patients? Did the patients know they had a MetS condition?

8.      Lines 184-185. How to test for the homogeneity of covariances by Box test. What types of box tests were used? Boxplot or something else?

9.      Lines 185-187. How to control for confounding effects produced by the covariates?

10.  Lines 245-253. The results obtained from the two-way MANCOVA were the main findings. The authors should make a table to present the results. Unfortunately, the two-way MANCOVA was not enough to evaluate the study issue, since this method did not control for confounding factors and residual confounding effects.

11.  Lines 253-259. The univariate results shown in Table 3 did not taken into the confounding effects into account. These results are not suitable for the following research conclusions.

12.  Lines 375-379. These statements were about the treatment and prevention of MetS, these were not the questions to be investigated in this study.

13.  Lines 379-381. Same issue! these statements were not drawn from this study and they should be omitted in the conclusions.

Author Response

Thank you for your precious comments and feedback on our paper. We were able to address the points you´ve mentioned. Below you find the point-to-point response and all changes are highlighted in the text marked with track changes. In addition, our senior authors proof-red the manuscript again.

We would like to express our thanks for your time and the questions raised which have been helped us to improve our manuscript significantly.

Round 2

Reviewer 1 Report

1.      The methods should be written in logical sequence as the results were presented.

1)       L139–149 Clarify “this study” L139, This sub-analysis”. This analysis may be conducted using a sup-population having F32 and F33 out of psychiatric rehabilitation center patients. L365 “Since depression combined with MetS is associated with significantly higher depression scores, treating or preventing is an essential factor [18, 19].” According to the Table 1, however, depression patients without metabolic syndrome has higher prevalence of N32, and N33 status. This is not coincident with the Introduction either. All participants to this study have unipolar depression, F32 and F33. What is the percentage of in Table 1?

2)       L165 There are 78 items, 13 strategies of 6 items, and 3 aggregated strategies (positive, negative, and neutral), out of which one aggregated strategy (positive) is divided into 3 subscale strategies. But one of 3 aggregated strategies, Neutral one, was not used for analysis. Four strategies (three positive subscale, and one negative aggregated strategies) were used. This should be clearly described.

3)       L171 “SVF uses T-normed data.” The information in L201, “be standardized to a T score with a mean of 50 according to the SVF-78 manual [32] to classify the data in relation to a non-clinical population aged between 20-79. Is inserted here. The score of each strategies (3 positive, and negative strategies) for analysis was standardized as T score. The reference is a non-clinical population aged 20–79 years, and mean and standard deviation of the reference population correspond to 50 and 10 of T score respectively.

4)       L171 “therefor” typo.

5)       In statistical analysis, descriptive analysis is mentioned first, followed by simple test (chi-square test, t test), and multivariate analysis. Sample size calculation is put in the section of participants selection. Univariate results, and univariate statistics L245, L250 and Table 4 may be not univariate analysis, but they are ANCOVA with one dependent variable, two factors, and three covariates. Univariate analysis means one with one dependent and one independent variables.

2.      This reviewer cannot access Ref. 32. The authors often wrote “to use more strategy” in the Results and Discussion. What does mean “more strategies”? The absolute score may be comparable between groups, whether raw or standardized. But, rather than that, three scales (positive, negative and neutral) are in balance, that is, positive strategies are used more prominently than negative strategies in individual stress coping. In this case, the ratio of negative to positive strategies instead of absolute ones may be appropriate. For example, in  https://www.researchgate.net/publication/248399754, negative and positive coping strategies are competitive as the results of this study. The author should clarify the meaning of “using more strategies”.

1)     L228, L296 T score was used for description of the distribution of strategies. The explanation, “The mean T-scores of all coping strategies are within the normal range, indicating that the strategies are used were at an average level,” is not informative. For example, “mean T-score = 60” means 84% of the sample has scores above mean of the references (frequency, or degree is not known here) under the assumption of normal distributions. Manuscript should be revised. F values in MANCOVA analysis are not dependent whether raw or standardized score.

2)     L232 A “normal rage” is awkward. When the mean T score was 60, 14% of the sample may be “abnormal”. What does the score above 60 mean?

3)     L240 In MONCOVA analysis, coefficients of each factor (metabolic syndrome and sex), or unbiased (least squared) means of scores for each factor should be presented. Strategies scores are not independent each other. Scores of positive and negative strategies may be in inverse relation. It is possible that effects (coefficients, or unbiased means) would be different from the results of Table 1).

4)     L268 “Individuals with depression with MetS used more distraction strategies than individuals without MetS indicating that individuals with depression.” What does “to use more distraction strategies” mean? Frequency, degree, amount, or efficiency? To eat food more often? To eat more food amount? To eat appropriately?

5)     L268 What is “additional MetS”? Do number of items of metabolic syndrome criteria increase?

6)     L270 “Women with depression used more distraction strategies but also significantly more negative strategies than men.” Differentiate “use more” from “more negative”.

7)     L276 “one previous study” “the previous study”

8)     L291 “Individuals with depression used slightly more negative strategies than positive strategies when dealing with stress.” This should be clarified. more negative than positive? To use negative strategies more often/frequently/vigorously/effectively/ than positive strategies.

9)     L296 “Our result might indicate that MetS no longer has an impact on negative strategies in individuals with depression when negative strategies are already increasingly used. However further research is needed to validate this result.” These sentences are ambiguous. What the authors intended may be “using negative strategies saturated (or ceiling effect) at 60 of T scores”. Please revise to clarify this. The authors can stratify the sample into high and low negative strategies score and analyze them separately, because the authors examined the distribution of the scores. They don’t need to wait for a future study.

10) L300 “individuals with depression and additional MetS” means individuals with coexisting of depression and metabolic syndrome”?

11) L301 clarify “more distraction strategies”.

12) L309 “higher caloric and excessive cholesterol intake” Despite of terms used in Ref. 35, “high energy intake” or “high energy-density” is better. Cholesterol intake seldom influence weight gaining. “Cholesterol” is not hit in the Ref. 35.

13) L320 Clarify “more distraction strategies”.

14) L329 Clarify “women with depression used more negative strategies than male individuals with depression”.

Author Response

Dear Reviewer, 

Thank you for taking the time to review our manuscript one again. Your comments and suggestions have been extremely valuable and have helped us to improve the quality of our work.

In response to your concerns, we have carefully re-written some passages of the manuscript. We would like to thank you for bringing these issues to our attention and for helping us to strengthen the rigor of our research. Your feedback has been essential in ensuring that our manuscript meets the highest standards of scientific excellence.

Once again, we are grateful for your time and effort in reviewing our manuscript. We hope that our revised version will meet with your approval, and we look forward to hearing from you soon.

Sincerely, 

Nina Dalkner and collegues

Reviewer 4 Report

The revised manuscript has improved, however, it still did not clarify some important issues in the article. MetS is a risk state, not an apparent or definitive disease, it cannot induce behavior in subjects to develop coping strategy for stress. Therefore, it is most likely that some confounding factors influencing stress coping strategies coexisted with MetS, such as motivational patterns, ego resources, defensive personality, beliefs about the environment and personal resources, in the observed data. However, the authors did not take these issues into account, making the conclusions of this paper uninterpretable.

Author Response

Dear Reviewer, 

Thank you for taking the time to review our manuscript one again. Your comments and suggestions have been extremely valuable and have helped us to improve the quality of our work.

In response to your concerns, we have carefully re-written some passages of the manuscript and tried to clarify the aspects you have mentioned. We would like to thank you for bringing these issues to our attention and for helping us to strengthen the rigor of our research. Your feedback has been essential in ensuring that our manuscript meets the highest standards of scientific excellence.

Once again, we are grateful for your time and effort in reviewing our manuscript. We hope that our revised version will meet with your approval, and we look forward to hearing from you soon.

Sincerely, 

Nina Dalkner and collegues

Round 3

Reviewer 1 Report

The authors mostly responded to the comments. The authors should address two comments further.

The authors added a limitation in Line 399; “There is a potential inverse dependence between negative and positive strategies, that may result in reduced statistical power.” The authors can analyze the data using inverse negative-strategies score and check whether the results change. A T score of negative strategies (Tneg) is easily transformed to an inverse T score as 100 minus Tneg, and the transformed T score instead of the original one is included in the MANCOVA model. After checking this, the authors should diagnose whether this is a limitation, or this reinforces the results. In this procedure, it is necessary to add methods and results.

Line 222-225 is a repetition of Line 181-184. “Furthermore, the score of each strategy 222 (3 positive, and negative strategy) was standardized as T score. The reference is a non- 223 clinical population aged 20–79 years, and mean and standard deviation of the reference 224 population correspond to 50 and 10 of T score respectively [32].”

Author Response

Dear Reviewer, 

thanks, that was a good idea. Consequently, we have done as suggested (using invers Tneg score in the MANCOVA model), but the results did not change significantly after using the inverse T score.

Results:

The two-way MANCOVA showed after using the inverse Tneg still a significant main effect MetS [F(4,353) = 2.771, p < 0.05, ηp2 = 0.030, Wilks’ Λ = 0.970] and a significant main effect sex [F(4,353) = 6.501, p < 0.001, ηp2 = 0.069, Wilks’ Λ = 0.931]. The interaction MetS by sex on the combined dependent variables was still not significant [F(4,353) = 1.405,  p = 0.232, ηp2 = 0.016, Wilks’ Λ = 0.984].

ANCOVAs showed that MetS groups (with vs without) differed still in distraction strategies [F(1,356) = 7.634, p < 0.05, ηp2 = 0.021] . In addition, there was still a statistically significant difference between sex (female vs male) for re-& devaluation strategies (Pos 1) [F(1,356) = 3.912, p < 0.05, ηp2 = 0.011], distraction strategies (Pos 2) [F(1,356) = 12.260, p < 0.001, ηp2 = 0.033], and negative strategies [F(1,356) = 21.190, p < 0.01, ηp2 = 0.025]. The significance in the negative strategies changed from p<0.001 to p<0.01.

We revised the limitations:

“There might be a potential inverse dependence between negative and positive strategies, that may result in reduced statistical power. However, we have checked that by including also inverse T-scores into the MANCOVA model indicating no change of results.”

Thank you also for your precise reading, we deleted the sentence in line 222-225.

Reviewer 4 Report

None.

Author Response

Thank you very much for your time.